# netboxr: Automated discovery of biological process modules by network analysis in R

**Eric Minwei Liu**[1]* , **Augustin Luna**[2,3,4]* , **Guanlan Dong**[5], **Chris Sander**[2,3,4]*

**1** Department of Biostatistics and Epidemiology, Memorial Sloan Kettering Cancer Center, New York, NY, United States of America, **2** Department of Cell Biology, Harvard Medical School, Boston, MA, United States of America, **3** cBio Center, Department of Data Sciences, Dana-Farber Cancer Institute, Boston, MA, United States of America, **4** Broad Institute of MIT and Harvard, Cambridge, MA, United States of America, **5** Department of Biomedical Informatics, Harvard Medical School, Boston, MA, United States of America

☯ These authors contributed equally to this work.

\* lium2@mskcc.org (EML); aluna@jimmy.harvard.edu (AL); sander.research@gmail.com (CS)

## Abstract

### Summary

Large-scale sequencing projects, such as The Cancer Genome Atlas (TCGA) and the International Cancer Genome Consortium (ICGC), have generated high throughput sequencing and molecular profiling data sets, but it is still challenging to identify potentially causal changes in cellular processes in cancer as well as in other diseases in an automated fashion. We developed the netboxr package written in the R programming language, which makes use of the NetBox algorithm to identify candidate cancer-related functional modules. The algorithm makes use of a data-driven, network-based approach that combines prior knowledge with a network clustering algorithm, obviating the need for and the limitation of independently curated functionally labeled gene sets. The method can combine multiple data types, such as mutations and copy number alterations, leading to more reliable identification of functional modules. We make the tool available in the Bioconductor R ecosystem for applications in cancer research and cell biology.

### Availability and implementation

The netboxr package is free and open-sourced under the GNU GPL-3 license R package available at https://www.bioconductor.org/packages/release/bioc/html/netboxr.html

## Introduction

Large-scale sequencing consortia such as The Cancer Genome Atlas (TCGA) [1] and the Interactional Cancer Genome Consortium (ICGC) [2] provide detailed genomic alteration profiling in many cancer types. Many methods based on the recurrence of genomic alterations, i.e., the frequency of occurrence in sets of tumor samples, have been developed to identify alterations likely to be functional in oncogenesis or cancer progression, addressing an important question in the field of precision oncology [3]. However, due to the considerable patient-to-

**Data Availability Statement:** The data underlying the results presented in the study are available from the netboxr package found at https://doi.org/10.18129/B9.bioc.netboxr.

**Funding:** This research was supported by the US National Institutes of Health grant (U41 HG006623-02), the Ruth L. Kirschstein National Research Service Award (F32 CA192901), and through funding for the National Resource for Network Biology (NRNB) from the National Institute of General Medical Sciences (NIGMS -P41 GM103504). The funders had no role in study design, data collection and analysis, decision to publish, or preparation of the manuscript.

**Competing interests:** The authors have declared that no competing interests exist.

patient heterogeneity of the cancer genome, rare mutations in certain patients can still be involved in tumor development by affecting biological processes in ways similar to those of known cancer genes. One way to address the issue of the effect of rare mutations is to combine prior knowledge of genetic and molecular interactions with recurrence-based methods and thus increase the power of predictions despite relatively low recurrence counts. In this spirit, we have developed the NetBox algorithm that seeks to automate the identification of candidate oncogenic processes and involved genes, which allows the quantitative analysis of genomic alterations in the context of known signaling pathway connectivity [4]. The NetBox algorithm identifies potentially novel network modules by mapping genomic alterations onto a comprehensive prior-knowledge interaction network, containing nodes and their interactions (edges), and then identifying modules as clusters of connected nodes that are frequently affected by genomic alterations as a set. The aggregation of nodes into clusters overcomes the statistical problem of low counts for individual nodes. This is in contrast to methods such as gene set enrichment analysis (GSEA), a popular approach to associate a gene list to biological functions, that relies on curated, pre-defined clusters of genes and does not make use of the often known interactions between genes or gene products. Unlike GSEA, NetBox is not limited to nor influenced by curated gene sets for module discovery and overcomes the issue of the occurrence of genes in more than one of the curated gene sets (Fig 1). Instead, the NetBox algorithm derives network modules de novo, based on the alteration data in tumor samples, such that the identified modules can identify new functional gene groups that cross the boundaries of curated gene sets. For newly discovered modules, the functional annotation can then be derived from the annotation of the gene members, providing potentially novel hypotheses about cellular processes that matter in the system from which the alteration data is derived.

To extend the use of NetBox, we have implemented the NetBox algorithm as a native R package, netboxr. The netboxr package provides users with access to the NetBox algorithm within the R ecosystem, thereby providing simplicity and flexibility for the visualization and secondary analyses through available R packages by using common data structures in R packages. Here, we describe the use of netboxr to integrate various types of genomic alterations for the detection of potentially functional network modules in glioblastoma multiforme (GBM) cancer as an example and highlight netboxr tutorial material for integrating the use of the netboxr package with additional R packages.

## Methods

### Implementation

The netboxr package implements the original NetBox algorithm for the discovery of pathway modules [4] using the R programming language and adds several functions to communicate with other packages in the R ecosystem and to integrate several input data types.

### Base functionality and algorithm

netboxr takes genes that are significantly altered by mutations, copy number alteration associated with gene expression change, or possibly changes in other data types, as input for identification of pathway modules. As a first step in the analysis, an input group of altered genes is mapped onto an interaction network from a comprehensive knowledge base of interactions; sources for networks of interactions include Pathway Commons using the paxtoolsr package. In NetBox, to account for obviously incomplete knowledge, candidate linker genes are defined as genes that do not have alterations but are direct neighbors of altered genes. To do this, in the second step, a hypergeometric test is used to determine the probability that a given candidate linker node has x or more interactions with nodes in the input gene list, $Pr(X{\geq}x)$, where x

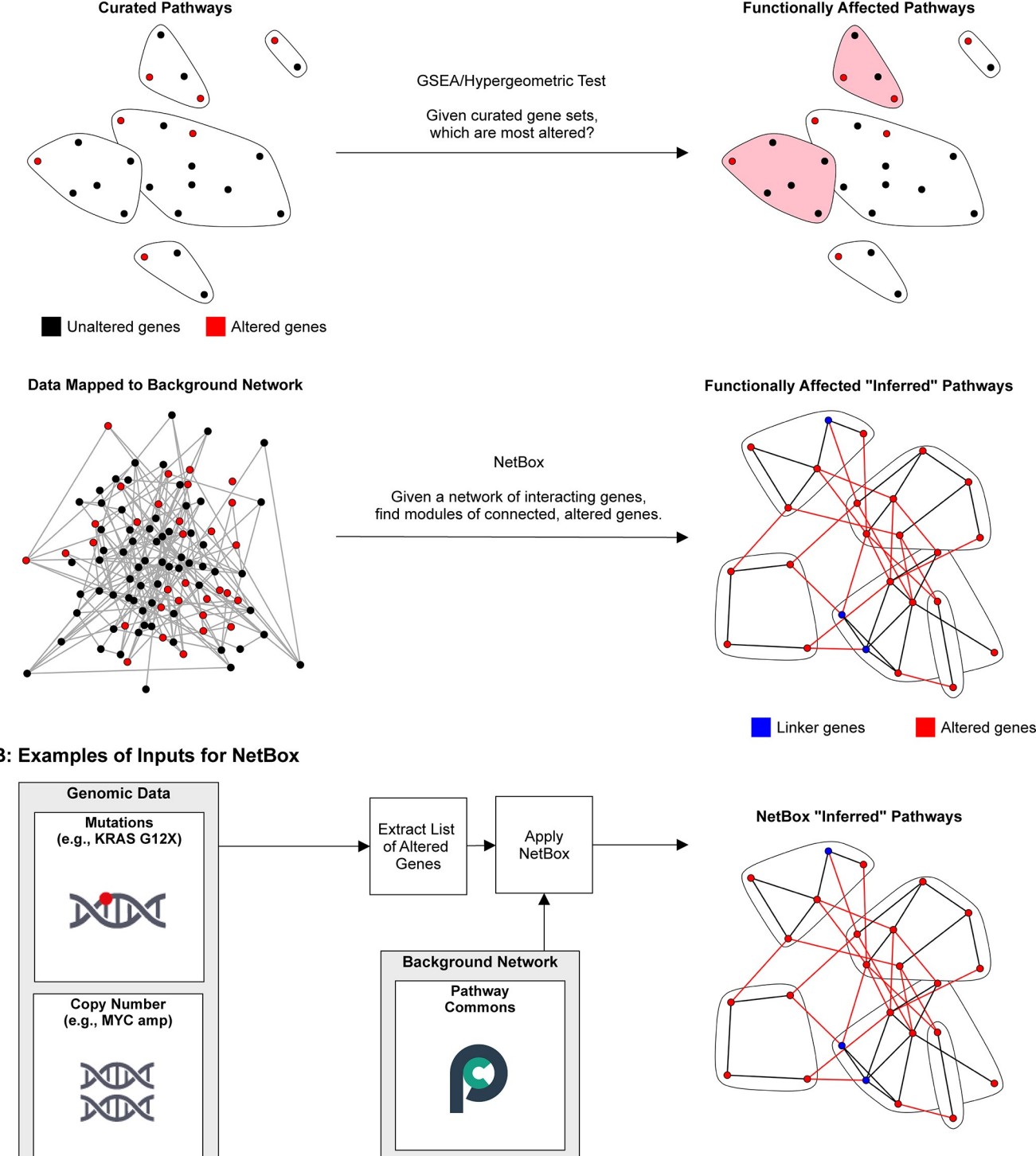

**Fig 1. Overview of NetBox algorithm.** A) The key difference between gene set enrichment analysis (GSEA)/hypergeometric tests and NetBox pathway discovery. Nodes are genes or gene products (red: altered in the dataset; black: unaltered). Edges are known interactions. Gene sets (top) typically do not contain interactions and if they do, these are not used in GSEA. Interactions are explicitly used in NetBox to infer functional modules. B) NetBox workflow for, e.g., alterations in cancer genomics data. Other types of data, such as mRNA expression profiles and proteomics profiles, can be equally accommodated as input.

is the observed number of interactions between a candidate linker node to altered genes in the input list. This probability is taken as the p-value; significant p-values indicate that candidate linker genes are involved in the relevant biological processes along with the input genes. P-values of each candidate linker gene are corrected by the Benjamini-Hochberg method. Linker genes with an adjusted p-value equal to or less than 0.05 are counted as significantly connected linker genes. Third, a community detection algorithm (e.g., the Girvan–Newman edge betweenness algorithm) is applied to the extended network with connected altered genes and linker genes. Finally, netboxr offers edge-betweenness and leading eigenvector (for networks with large numbers of nodes) algorithms to identify network modules as connected clusters of genes. As newly configured modules lack functional labels, the modules identified by netboxr can then be passed to enrichment packages, such as the ClusterProfiler package, to characterize the modules in terms of functional Gene Ontology (GO) terms (i.e., biological processes, molecular function, or cellular compartment), to complement the functional annotation of individual genes with an overall functional label for the set of interacting genes, which we call pathway modules.

## Assessment of statistical significance

Two statistical tests are performed on the identified network modules to assess the significance of the identified network (i.e., the identified network is the entirety of the network defined by the altered gene nodes, linker nodes, and their connecting edges). These tests were conducted in a similar manner as for the original NetBox algorithm [4]. To assess the level of global connectivity, an empirical p-value is calculated by determining the number of times the size of the largest connected component (the largest network component can be composed of multiple modules) identified from the same number of randomly selected genes equals or exceeds the size of the largest connected component from the list of altered genes in the data set. Next, a network modularity score is calculated [5]. This score represents the strength (or quality) of the division of a network into various modules and is defined as the edge fraction that is within given modules minus an expected fraction by randomly distributing the edges. To assess the statistical significance of the network modularity observed in the resulting network, we used a local rewiring algorithm where random networks are generated that maintain the same size and all genes maintain the same degree, but the choice of interaction partners is random. For each of these random networks, we calculate the network modularity score and calculate the average and standard deviation for a set of random networks. The observed modularity score is then converted into a z-score (and reported as a p-value) to measure the deviation of the observed network modularity from that of the random null model.

## Implementation details and integration in the R ecosystem

Beyond the base functions, netboxr includes several additions to simplify and expand its use. These include 1) instructions for retrieving and processing genomic alterations such as mutations and copy number alterations from data repositories such as cBioPortal [6, 7] or Genomic Data Commons (GDC) [3], 2) instructions to use pathway data (i.e., genes or gene products with interactions) via the paxtoolsr Pathway Commons package [8] or from resources such as the STRING pathway database [9], 3) functionality to switch between the discovery of modules using various algorithms for detection of network communities from the igraph package [10] and 4) guidance on functionally annotating netboxr-derived modules using the ClusterProfiler package [11]. netboxr was implemented in R and can be installed through the BiocManager package manager from Bioconductor (bioconductor.org).

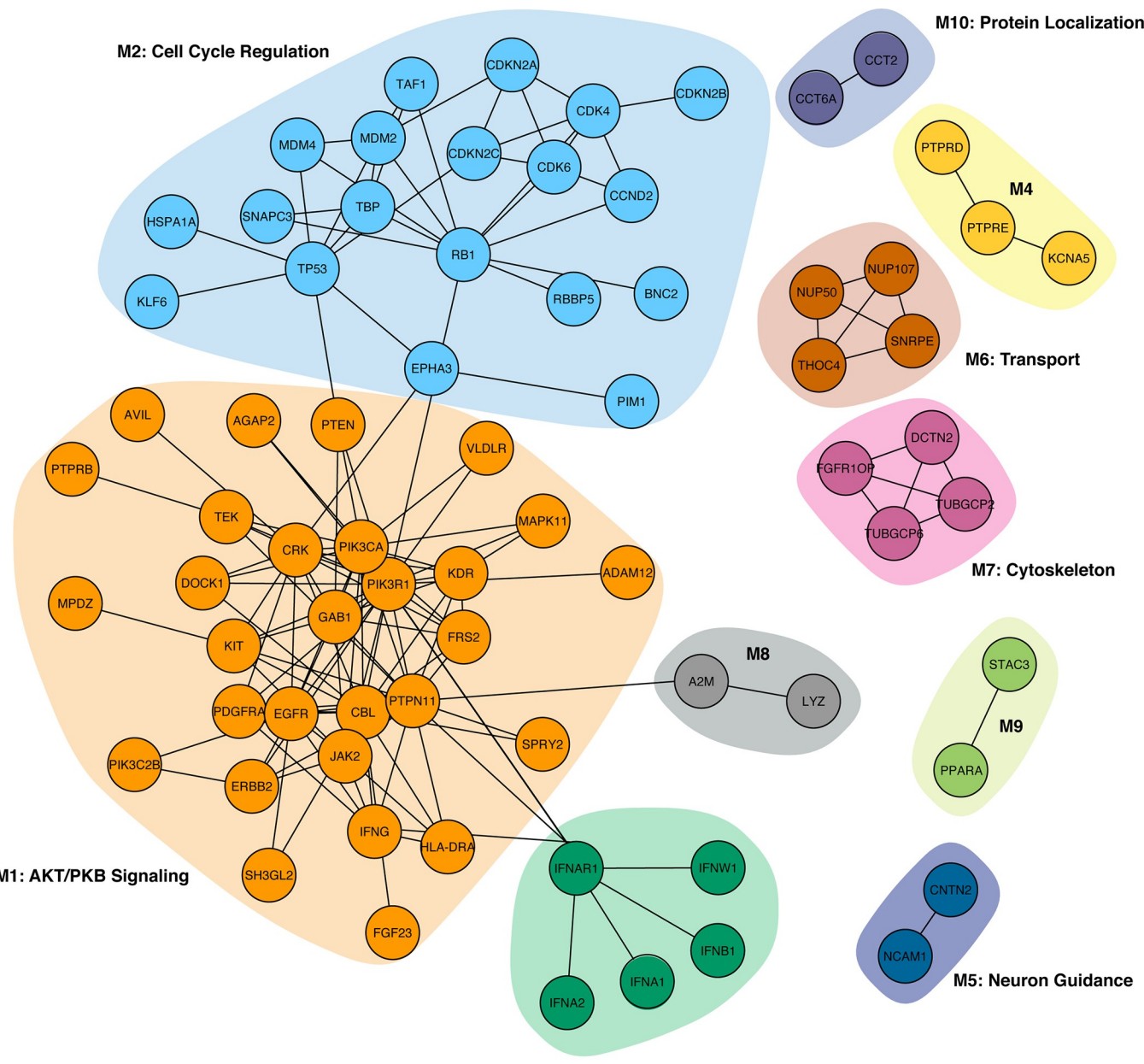

**Fig 2. Glioblastoma multiforme (GBM) pathway modules identified by the netboxr package from cancer genomics alteration data without the use of pre-defined gene sets.** The new NetBox algorithm implementation in R uses the igraph library to speed up module detection and visualization. Using mutations, and copy number alteration data from TCGA (Cerami et al., 2010), netboxr identified 10 pathway modules. Modules are functionally annotated with a brief description of the module genes using the clusterProfiler Bioconductor package (S1 File). For more detailed understanding users should also inspect the function of the genes contained in the modules. In this glioblastoma example, the largest module (M1, light orange background) contains genes related to the PIK3 pathway and functions related to AKT signaling (also known as "PKB signaling" in the GO Gene Ontology). The second-largest module (light blue background) contains genes related to the TP53 and cell cycle pathways. These algorithmically inferred two main modules are consistent with the intuitively inferred signaling pathways in the original TCGA publication [12].

## Use case

The package vignette (i.e., tutorial) provides step by step instructions for the usage of the package and exploration of the results. Additionally, within the vignette, we provide instructions

for users to generate input data and retrieve input gene lists from publicly available studies on cBioPortal [6, 7].

As a specific use case, here we tested the netboxr package in a cancer use case by using the list of altered genes from TCGA datasets and the prior-knowledge network data from the original NetBox paper [4]. The results reported here are comparable to those from Cerami et al. although the unadjusted p-values for linker genes are not exactly the same. This is because the unadjusted p-values of linker genes in the original NetBox report [4] were calculated as the probability that a given candidate linker node has exactly X interactions $Pr(X = x)$, where X is the observed number of interactions between a candidate linker node to altered genes in the input list; instead, netboxr uses a hypergeometric test as described in the Base Functionality section. The final number of linker genes, using a significance cutoff of 0.05 after FDR correction, is the same in netboxr and the original NetBox implementation. Using netboxr, we identify the PIK3R1 (Module M1) and RB1 (Module M2) modules, each with connected genes that are significantly altered as a set in the glioblastoma (GBM) cancer genomics data from the Cancer Genome Atlas (TCGA). Each of these modules is then annotated with brief descriptions through an enrichment step using the clusterProfiler Bioconductor package using gene annotations from the Gene Ontology (Fig 2; S1 File). Module M1 is related to AKT/PKB signaling while M2 is related to cell cycle regulation; brief descriptions of functions for other modules are shown in the figure. This module-driven exploration of the identified network allows a finer-grain understanding of the input gene list than through simply performing an enrichment analysis over the entire input gene list through 1) the topology of the identified network connections and 2) the annotation of specific modules (Fig 2, S1 File). Details for this example are provided in the vignette document in the netboxr package. Fig 2 demonstrates the use of the netboxr package to discover pathway modules from multiple genomic data for the example of the TCGA glioblastoma multiforme (GBM) study.

## Conclusion

The netboxr R package facilitates data-driven network module discovery. Here we provide a use case that emphasizes analysis of cancer genomics data, but the methodology is applicable for other diseases or cell biological perturbation experiments with comparable large datasets covering genetic or molecular changes in genes or gene products. With the ease of installation in R bioinformatics environments, researchers can quickly use datasets of molecular measurements to identify pathway modules and form biological hypotheses on the functional role of cellular processes in cancer and other diseases, as well as in healthy tissues. As with any method, users should carefully consider the problems they attempt to address and understand the strengths and limitations before using a particular method. As for the contrast with enrichment analyses, which NetBox is distinct from and complementary to, Reimand et al. present a broader discussion of potential biases, including those in existing pathway databases [13]. The netboxr authors can be contacted to explore collaboration in module discovery in large scale perturbation-response datasets of interest.

## Supporting information

**S1 File.**
(DOCX)

## Acknowledgments

We would like to thank The Cancer Genome Atlas community of researchers for providing the genomic data used in the netboxr package vignette.

## Author Contributions

**Conceptualization:** Chris Sander.

**Formal analysis:** Eric Minwei Liu.

**Funding acquisition:** Augustin Luna, Chris Sander.

**Methodology:** Eric Minwei Liu, Augustin Luna.

**Project administration:** Chris Sander.

**Software:** Eric Minwei Liu, Augustin Luna, Guanlan Dong.

**Visualization:** Eric Minwei Liu, Guanlan Dong.

**Writing – original draft:** Eric Minwei Liu, Augustin Luna.

**Writing – review & editing:** Augustin Luna, Guanlan Dong, Chris Sander.

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
