## [Decision Letter · Decision Letter 0]

23 Jul 2020

PONE-D-20-16149

NetBoxR: Automated Discovery of Biological Process Modules by Network Analysis in R

PLOS ONE

Dear Dr. Luna,

Thank you for submitting your manuscript to PLOS ONE. After careful consideration, we feel that it has merit but does not fully meet PLOS ONE’s publication criteria as it currently stands. Therefore, we invite you to submit a revised version of the manuscript that addresses the points raised during the review process.

We look forward to receiving your revised manuscript.

Kind regards,

Tao Huang

Academic Editor

PLOS ONE

Journal Requirements:

2. During our initial internal evaluation of your submission we noticed several typos in your manucript.

Please note that PLOS ONE does not providing copyediting or proofs of accepted manuscripts.

We therefore recommend that you carefully review your manuscript and correct any errors at this time.

'Funding

This research was supported by the US National Institutes of Health grant (U41 HG006623-02),

the Ruth L. Kirschstein National Research Service Award (F32 CA192901), and through funding

for the National Resource for Network Biology (NRNB) from the National Institute of General

Medical Sciences (NIGMS -P41 GM103504).'

'The funders had no role in study design, data collection and analysis, decision to publish, or preparation of the manuscript.'

Reviewers' comments:

Reviewer's Responses to Questions

**Comments to the Author**

1. Is the manuscript technically sound, and do the data support the conclusions?

Reviewer #1: Partly

Reviewer #2: Yes

2. Has the statistical analysis been performed appropriately and rigorously? 

Reviewer #1: Yes

Reviewer #2: Yes

3. Have the authors made all data underlying the findings in their manuscript fully available?

Reviewer #1: Yes

Reviewer #2: Yes

4. Is the manuscript presented in an intelligible fashion and written in standard English?

Reviewer #1: Yes

Reviewer #2: Yes

5. Review Comments to the Author

Reviewer #1: The authors developed an R package named NetBoxR that makes use of the NetBox algorithm to identify candidate cancer-related processes. The algorithm makes use of a network kbased approach that combines prior knowledge with a network clustering algorithm. This program can combine multiple data types, such as mutations and copy number alterations, leading to more reliable identification of functional modules. This tool can be useful for the research community to identify the functional modules in cancer data analysis.

I have some little concerns:

1. In the paper, it says” NetBoxR takes significantly altered genes from SNV mutation, copy number alteration associated with gene expression change, and merges them into a gene list for identification of pathway modules.” How can users to define the “significantly altered genes from SNV mutation, copy number alteration”, just like finding differentially expressed genes from expression profiles, how to take the variants into account to find these genes? Authors may provide some guidance on choosing these genes for a better performance in NetBoxR.

2. From the methodology side, there are differences between the GSEA algorithm and NetBox, it seems the NetBox overcomes some shortcomes, I suggest authors to do a comparison between the results from GSEA and NetBox, to provide the new findings from NetBox, which can not be got from GSEA.

3. In the online tutorial, the results from NetBox were used for GO enrichment, why not to do the GO enrichment at the first using the original gene list, what are the advantages by using the NetBoxR to do a pre-process before the GO enrichment?

4. In the result figure (paper, online), it shows some modules, but there is no annotation of these module? The author should show the function annotation for the modules in the result figures found by their program.

5. Minor: the text in original figure can be see clearly, but I cannot see any word in the figure embedded in the pdf file.

Reviewer #2: Liu et al present the R package NetBoxR that implements the NetBox algorithm previously described in an article by Cerami et al (2010). This article presents little in terms of new methods, data or results; the main message is the implementation of the NetBox algorithm in R rather than a standalone command-line application.

The article is mostly written well and is easy to understand, at least if the reader is willing to refer to the Cerami et al work for some of the details. There are some sentences that could use rewording to make them clearer, see below.

In terms of methods, I am concerned mainly about two issues:

1. NetBox and NetBoxR evaluate the significance of the found network by focusing on the largest module only. It would be beneficial to implement a method that can determine significance of each module, not just the largest one.

2. The identified modules are based on literature networks, which presumably makes any enrichment analysis in literature sets biased. If I understood the methods correctly, this inflation is the result of not just the connections that come from the literature, but also because linker genes are derived from the literature and included in the modules. It does not appear that the package contains any functions that could quantify and correct the bias.

As for implementation, a cursory reading of the help files indicates that they need be improved. Reading the help for geneConnector, one learns for example that the argument networkGraph is “an igraph graph object” but it’s unclear what it is supposed to represent. Similarly, argument communityMethod is described as “A string for community detection method c('ebc','lec')”. It is not clear what the shorthands mean. Arguments ‘directed’ and ‘keepIsolatedNodes’, although both logical, are described as “TRUE of FALSE” and “logic value”, respectively; neither has any information on what the TRUE and FALSE values actually determine. Similar comments apply to the description of the output of the function. The individual input arguments and output components need a description of what they represent, not just what type of R object is expected.

The authors say that the package provides “instructions for retrieving and processing genomic alterations such as SNV mutations and copy number alterations from data repositories such as cBioPortal (Cerami et al., 2012; Gao et al., 2013) or Genomic Data Commons (GDC) (Jensen et al., 2017)”. I did not find the instructions; perhaps I missed something in the vignette or in the help files?

Minor issues:

The authors refer to the package as NetBoxR, but the package name on bioconductor is netboxr (all lowercase). This can cause confusion, e.g., NetworkManager::install(“NetBoxR”) produces an error.

The sentence “ability to retrieve pathway data via the paxtoolsr package as available in ~20 data source aggregation Pathway Commons, (Luna et al., 2016) or STRING (Snel et al., 2000)” does not make sense.

Contrary to what the authors seem to imply, the package does not seem to be available for R-3.6.3; one apparently needs R-4.0.0 or higher.

6. PLOS authors have the option to publish the peer review history of their article (what does this mean?). If published, this will include your full peer review and any attached files.

Reviewer #1: No

Reviewer #2: No

---

## [Author Response · Author response to Decision Letter 0]

21 Sep 2020

Thanks for the constructive feedback. We revised our manuscript accordingly. Please see our response as a separate document.

---

## [Decision Letter · Decision Letter 1]

14 Oct 2020

netboxr: Automated discovery of biological process modules by network analysis in R

PONE-D-20-16149R1

Dear Dr. Luna,

We’re pleased to inform you that your manuscript has been judged scientifically suitable for publication and will be formally accepted for publication once it meets all outstanding technical requirements.

Kind regards,

Tao Huang

Academic Editor

PLOS ONE

Additional Editor Comments (optional):

Reviewers' comments:

Reviewer's Responses to Questions

**Comments to the Author**

1. If the authors have adequately addressed your comments raised in a previous round of review and you feel that this manuscript is now acceptable for publication, you may indicate that here to bypass the “Comments to the Author” section, enter your conflict of interest statement in the “Confidential to Editor” section, and submit your "Accept" recommendation.

Reviewer #1: All comments have been addressed

Reviewer #2: All comments have been addressed

2. Is the manuscript technically sound, and do the data support the conclusions?

Reviewer #1: Yes

Reviewer #2: Yes

3. Has the statistical analysis been performed appropriately and rigorously? 

Reviewer #1: Yes

Reviewer #2: Yes

4. Have the authors made all data underlying the findings in their manuscript fully available?

Reviewer #1: Yes

Reviewer #2: Yes

5. Is the manuscript presented in an intelligible fashion and written in standard English?

Reviewer #1: Yes

Reviewer #2: Yes

6. Review Comments to the Author

Reviewer #1: I did not find the response to the comments file in the review system(I am not sure if the authors forgot to submit it or something wrong with the review system).

I have read through the paper, and I can feel that authors has made great upgrade to their netboxr package, and refined the manuscript. It can be accepted.

One minor comment: There are too much kay words.

Reviewer #2: (No Response)

7. PLOS authors have the option to publish the peer review history of their article (what does this mean?). If published, this will include your full peer review and any attached files.

Reviewer #1: No

Reviewer #2: No

---

## [Editor Report · Acceptance letter]

22 Oct 2020

PONE-D-20-16149R1 

netboxr: Automated discovery of biological process modules by network analysis in R 

Dear Dr. Luna:

I'm pleased to inform you that your manuscript has been deemed suitable for publication in PLOS ONE. Congratulations! Your manuscript is now with our production department. 

Kind regards, 

on behalf of

Dr. Tao Huang 

Academic Editor

PLOS ONE